# Adsorptive Removal of Arsenic and Lead by Stone Powder/Chitosan/Maghemite Composite Beads

**DOI:** 10.3390/ijerph18168808

**Published:** 2021-08-20

**Authors:** Hun Pak, Jesse Phiri, Junhyung We, Kyungho Jung, Sanghwa Oh

**Affiliations:** School of Architectural, Civil, Environmental, and Energy Engineering, Kyungpook National University, Daegu 41566, Korea; ph6524@knu.ac.kr (H.P.); jesset.phiri@knu.ac.kr (J.P.); gud6417@knu.ac.kr (J.W.); nabi6030@daum.net (K.J.)

**Keywords:** arsenic, lead, maghemite, chitosan, stone powder, bead, adsorption

## Abstract

Arsenic (As) and lead (Pb) contamination in groundwater is a serious problem in countries that use groundwater as drinking water. In this study, composite beads, called SCM beads, synthesized using stone powder (SP), chitosan (Ch), and maghemite (Mag) with different weight ratios (1/1/0.1, 1/1/0.3, and 1/1/0.5 for SP/Ch/Mag) were prepared, characterized and used as adsorbents for the removal of As and Pb from artificially contaminated water samples. Adsorption isotherm experiments of As and Pb onto the beads were conducted and single-solute adsorption isotherm models such as the Langmuir, Freundlich, Dubinin–Radushkevich (DR), and dual mode (DM) models were fitted to the experimental data to analyze the adsorption characteristics. The maximum adsorption capacities of the SCM beads were 75.7 and 232.8 mmol/kg for As and Pb, respectively, which were 40 and 5.6 times higher than that of SP according to the Langmuir model analyses. However, the DM model had the highest determinant coefficient (R^2^) values for both As and Pb adsorption, indicating that the beads had heterogenous adsorption sites with different adsorption affinities. These magnetic beads could be utilized to treat contaminated groundwater.

## 1. Introduction

Groundwater contamination has become a serious environmental concern throughout the world. In several countries such as Bangladesh, Vietnam, Pakistan, and China as well as the USA, Hungary, and Mexico, many people are exposed to high levels of arsenic (As) and/or lead (Pb) via the intake of contaminated groundwater [1,2,3,4,5]. As and Pb are considered to be the most hazardous chemicals in the world, and the excessive and long-term intake of these chemicals causes several serious health problems, including suffering from incurable diseases, such as cancer, and potentially causing death [6]. In particular, the high As concentration in the groundwater in southeastern Asian countries poses a serious threat to the lives of many people.

Various technologies for As and Pb removal from contaminated water have been developed, including adsorption, ion exchange, pH control, and precipitation [7,8]. Adsorption is considered to be one of the more attractive technologies, and various adsorbents that are effective in As and Pb removal have been developed. For As removal, iron-based compounds such as goethite [9,10,11], hematite [11,12], magnetite [11,12], maghemite [12,13], ferrihydrite [12,14], zero valent iron [11,12,15,16], amorphous hydrous ferric oxides [16,17], iron-coated materials [12,18], and mackinawite [19], have been considered as preferred adsorbents because iron oxides are abundant in nature. For Pb removal, soils [20,21,22,23], zeolite [24], carbon aerogel [25], and metal oxides, such as iron oxides [26,27,28] and aluminum oxides [29], have been studied.

Iron oxides have excellent adsorption capacities for As and Pb. Ghosh et al. [9] reported that goethite nanoparticles have a high As adsorption capacity (76 mg/g). Tuutijarvi et al. [13] reported that maghemite nanoparticles adsorb high amounts of As (50 mg/g). However, because iron oxide usually exists as a fine powder, it is difficult to apply in field treatments [30]. Therefore, to improve its field applicability, magnetic (nano) particles synthesized from iron oxide were researched and developed by several groups [31,32,33,34]. For example, Cho et al. [31] synthesized a chitosan/clay/Fe_3_O_4_ composite material to remove cationic and anionic dyes. Podder and Majumder [32] applied a granular activated carbon/MnFe_2_O_4_ composite to remove As. In addition, many studies have used chitosan as a cross-linker for the synthesis of composite beads, which can bridge and aggregate chitosan, iron-oxide and other powders [31,33,34]. Adsorbents with magnetic beads with biopolymers are attractive because of their adsorption efficiency and magnetic properties [34].

Chitosan-based adsorbents are interesting because of their eco-friendly properties [31]. Because chitosan contains hydroxyl (-OH) and amino (-NH_2_) groups, it has a problem with liquidation in which its structure collapses under acidic conditions. A cross-linking agent, such as triphosphate (TPP), can be used to prevent liquidation [35]. Cho et al. [31] and Ngah and Fatinathan [36] reported that TPP bridges chitosan polymers with protonated amines and encapsulates it into a bead form. For example, Cho et al. [31] synthesized a chitosan/clay/Fe_3_O_4_ (CCM) composite in which chitosan acts as a cross-linker between the clay and the Fe_3_O_4_ and the clay acts as a frame structure. In this study, stone powder (SP) was used to act as a frame structure instead of clay. Untreated SP waste can cause environmental pollution in soil, surface water, and groundwater. Only a small portion of SP is recycled for construction purposes [37] but it is not used for environmental purposes because of its poor manageability.

This study, accordingly, synthesizes a bead containing maghemite, chitosan, and SP for the adsorption of As and Pb and evaluates the performance of the composite bead adsorbent and the factors affecting adsorption, for example, the maghemite/SP weight ratio, bead dose, and temperature. Several adsorption isotherm models, i.e., the Langmuir, Freundlich, Dubnin-Radushkevich (D-R), and dual-mode (DM) models were used to fit the adsorption data.

## 2. Materials and Methods

### 2.1. Materials

SP was obtained from a masonry mill in Yeongcheon, Korea, air-dried, sieved using a 75 μm mesh (sieve #200), and stored in an airtight plastic bottle before use [38]. Chitosan ((C_6_H_11_NO_4_)_n_ with 75–85% degree deacetylation and a viscosity (0.5%) of 5–20 mPa∙s) was purchased from Showa, Japan. Sodium arsenate dibasic heptahydrate (Na_2_HAsO_4_, >99.0%) was purchased from Wako, Japan. Lead nitrate (Pb(NO_3_)_2_, >99.0%), acetic acid (CH_3_COOH, >98.0%), iron(II) chloride (FeCl_2_, >99.0%), iron(III) chloride anhydrous (FeCl_3_, >98.0%), and hydrochloric acid (HCl, 35–37%) were purchased from Duksan Co., Ansan, Korea. MES (2-(*N*-morpholino)ethanesulfonic acid, >98.0%), sodium tripolyphosphate (STPP, NaP_3_O_10_, >99.0%), and sodium bicarbonate (NaHCO_3_, >99.0%) were purchased from Daejung Chem. & Metals Co., Siheung, Korea. Iron(III) nitrate (Fe(NO_3_)_3_ 9H_2_O, >98.0%) was purchased from OCI Co., Seoul, Korea.

### 2.2. Ferrofluid Synthesis

A ferrofluid composed of maghemite (γ-Fe_2_O_3_) was synthesized according to the modification of the Massart method [39]. In brief, a 0.7 M ammonia solution (NH_4_OH) was added to a mixture of iron(III) chloride (40 mL, 1 M) and iron(II) chloride (10 mL, 2M) to form black-colored settled particles. The particles were separated via centrifugation at 1500 rpm for 10 min, dispersed into a 0.34 M nitrate solution with a pH of 2, and then oxidized to maghemite with iron(III) nitrate at 90 °C for over 1 h. The synthesized maghemite concentration in the ferrofluids was approximately 10 g/L as solids.

### 2.3. SP/Chitosan/Maghemite Bead Synthesis

The synthesis protocol for the SP/chitosan/maghemite (SCM) beads was based on the protocol of Bee et al. [34]. Briefly, 2 g of chitosan was dissolved in 100 mL of 2% acetic acid. Then, 2 g of SP and an appropriate amount of ferrofluid solution corresponding to the different weights of 0.2, 0.67, or 1.0 g of maghemite were added to the chitosan solution. The solution was then mixed at approximately 50 °C until the viscosity of the mixed solution reached approximately 25–28 mPa·s. In addition, 500 mL of 0.5 M sodium triphosphate (STPP) was prepared in a 1 L beaker. The SCM mixture solution was transferred into a 50 mL syringe installed on a syringe pump and then added dropwise to the STPP solution to form beads. The beads were cured for approximately 24 h, washed twice with ultrapure water, and then dried at 50 °C in an oven for 24 h. The synthesized beads had three weight ratios of SP, chitosan, and maghemite: 1/1/0.1, 1/1/0.3, and 1/1/0.5.

### 2.4. Characterization of the SCM Beads

The surface shape of the SCM beads was observed using a microscope (Zeiss, Axioplan 2 Imaging, Axiovert 200, Jena, Germany). A field emission scanning electron microscope (FE-SEM, SU8220, Hitachi, Japan) was used to observe the SCM morphology and an energy dispersive X-ray spectroscope (EDS, Horiba E-MAX EDS detector, Kyoto, Japan) was used to characterize the chemical compositions. Fourier transform infrared spectroscopy (FT-IR) was used for the characterization of the bead structural features. The surface area and pore size of the SCM beads were measured using the Brunauer–Emmett–Teller (BET) method (BET Quantachrome, Autosorb-iQ, Boynton Beach, FL, USA) via the N_2_ adsorption isotherm. The BET, microscope, and FE-SEM/EDS analyses were conducted at the Instrumental Analysis Center of Kyungpook National University, Korea.

### 2.5. Adsorption Isotherm Experiments

Isothermal adsorption tests were performed to evaluate the adsorption capacity of the SCM beads and SP to As and Pb. All experiments were performed in a 50 mL centrifuge tube (PE, SPL Pocheon, Korea). A total of 0.5 g of SCM beads or 1.0 g of SP powder was prepared in screw-cap conical tubes with an available volume of 50 mL, and then metal solutions (As: 0.013–1.33 mmol/L or Pb: 0.965–9.65 mmol/L) with a background electrolyte of 0.01 M NaNO_3_ were added. The tubes were capped tightly and shaken at 200 rpm for 24 h in an orbital shaker. After mixing, all tubes were centrifuged at 2000 rpm for 10 min. The supernatant was filtrated through a 0.2 μm membrane filter (cellulose nitrate membrane, Whatman). The As and Pb concentrations in the aqueous phases were analyzed using an inductively coupled plasma (ICP, Optima 2100 DV, PerkinElmer, Hägersten, Sweden). The pH of the electrolyte solution was in the range of 4–5 adjusted with 0.1 N HCl and 0.1 N NaOH. All experiments were conducted in duplicate.

The solid phase adsorbed amount, *q* (mmol/kg), was calculated using Equation (1):(1)$$q=\frac{(C_{0}-C)V}{W}$$
where *C*_0_ is the initial solute concentration (mmol/L), *C* is the residual solute concentration (mmol/L), *V* is the sample volume (L), and *W* is the weight of the adsorbent SCM (×10^−3^ kg).

## 3. Results

### 3.1. SCM Characteristics

The specific area, pore volume, pore size, and pH values of the SCM beads and SP are summarized in Table 1. Because the SP used in this study is the same as that used in the previous study [36], the SP properties were also the same. The surface area of the SCM beads was 0.543–0.834 m^2^/g, which was approximately a quarter of that of SP (2.782 m^2^/g). The pore volume of the beads was in the range of 0.0027–0.0047 cm^3^/g which was lower than that of SP, whereas the pore size of the beads (20.0–27.9 nm) was similar to that of SP (24.8 nm).

Microscopic images and scanning electron microscopic (SEM) images show the shapes of the SCM beads in Figure 1. As shown in Figure 1a–c, the beads are brownish, have a round shape, and become darker as the maghemite content increases from 0.1 to 0.5 wt.%. The beads are hollow spheres (Figure 1b), and the surface of the beads was pitted as a result of shrinkage during drying as shown in Figure 1d–f (SEM images).

Figure 2 shows the SEM images and EDS graphs for the beads and SP. Figure 2a–d present the SEM images for the SCM beads (1/1/0.1, 1/1/0.3, and 1/1/0.5) and SP, respectively. The EDS analytic results are summarized in Table 2. As shown in Figure 2, O, C, and P are prominent elements for the beads whereas Si is the largest component of SP. In Table 2, the Fe contents in beads 1/1/0.1, 1/1/0.3, and 1/1/0.5 are 0.81, 2.62, and 2.89%, respectively, which are higher than the Fe content in SP (0.23%). The high Fe content in the beads can provide sufficient adsorption sites as reported by Ghosh et al. [9], i.e., iron-based adsorbents have high arsenic adsorption capacities. As can be seen in Table 2, the Fe content of bead 1/1/0.5 (2.89%) was lower than expected, and was not significantly different from that of bead 1/1/0.3 (2.62%). If the adsorption of As is dependent on the Fe content, the two beads would be expected to have similar adsorption capacities.

FT-IR spectra for the bead only, the As adsorbed bead, and the Pb adsorbed bead are shown in Figure 3. Several bands were observed at 2,150, 1630, 1000, 780, and 580 cm^−1^ and a wide crest was observed in the region from 3700 to 3000 cm^−1^ for all beads. In this region, the absorptions were caused by the N–H and O–H bonds. The bands at 1000 and 780 cm^−1^ could be attributed to −C−O str and As−O−Fe stretching vibrations, respectively. 

### 3.2. Adsorption Characteristics of As onto the Beads

The experimental results on the adsorption of As and Pb onto the beads and SP are presented here. Based on the data, nonlinear relationships between C and *q* were observed for all adsorption isotherm experiments. The adsorption isotherms of As onto the SCM beads and SP were studied by fitting the experimental data to the representative isotherm models such as the Langmuir, Freundlich, D-R, and DM models, as shown in Table 3. 

Figure 4 shows the adsorption isotherm of As onto the SCM beads with three different ratios of SP/chitosan/maghemite (1/1/0.1, 1/1/0.3, and 1/1/0.5) compared with that onto SP. As shown in Figure 4, the As concentration (*q*_e_) adsorbed onto the beads significantly increased compared with that on SP and increased with the maghemite ratio in the beads (1/1/0.5 > 1/1/0.3 > 1/1/0.1). All adsorptions showed a mixed pattern of C- and L-type isotherms.

Table 4 shows a summary of the fitted model parameters, the determinant coefficients (R^2^), and the squared standard errors (SSE) for As adsorption onto the beads and SP. The R^2^ values of the As adsorption onto the beads and SP were all higher than 0.90, indicating that all isotherm models can generate a satisfactory fit to the data. Of the models, the Langmuir and DM models fitted the data more accurately than the others.

In the Langmuir model, the maximum adsorption capacity, *q*_mL_, increased as the maghemite content increased and there was little difference between the *q*_mL_ values for beads 1/1/0.3 and 1/1/0.5. In the same concentration range, the maximum adsorption capacity increased by 24 times for bead 1/1/0.1 (43.94 mmol/kg), 35 times for bead 1/1/0.3 (65.06 mmol/kg), and 41 times for bead 1/1/0.5 (75.74 mmol/kg) compared with SP (1.84 mmol/kg). However, the *b* value—adsorption affinity—did not show a consistent pattern.

In the Freundlich model, the R^2^ values for the SCM beads and SP were above 0.96 and 0.90, respectively. The *K*_F_ values were found to be 25, 47, and 44 in the SCM beads 1/1/0.1, 1/1/0.3, and 1/1/0.5, respectively, and the ratio of maghemite was similar at 0.3 or higher. In addition, because the *K*_F_ value was approximately 10 times higher for the beads than that for SP, it was found that the adsorption affinity of the beads was higher than that of SP. The N values were found to be in the range of 0.66–0.71, indicating nonlinear behavior.

Table 4 also shows the results of fitting with the D-R model, one of the pore volume filling models. Overall, the R^2^ value was higher than 0.96. Because As atoms are 119 pm in diameter and exist in the form of arsenate in water, most of the adsorption appears to have occurred in the pores and surfaces of the beads with pore sizes of approximately 20 nm or more. The *q*_mD_ values of beads 1/1/0.1, 1/1/0.3, and 1/1/0.5 were 28.07, 47.22, and 44.30 mmol/kg, respectively, and the *q*_mD_ values of beads 1/1/0.3 and 1/1/0.5 were similar. This value was approximately 25 times higher than the 1.883 mmol/kg measured for SP. Comparing *q*_mD_ in the D-R model and *q*_mL_ in the Langmuir model, *q*_mD_ was smaller than *q*_mL_. However, the *E* values are all less than 8 kJ/mol, indicating that physical adsorption had occurred.

In this study, the dual mode (DM) model, which is a model combining the Langmuir model and the linear model, was also applied to analyze whether monolayer or multi-layer adsorption occurred. Table 4 shows an R^2^.value of 0.97 or higher for the DM model, which was the same as that for the Langmuir model. The *q*_mDM_ value of the DM model and the *q*_mL_ value of the Langmuir model were very similar, and the *K*_pDM_ value was very low (less than 0.049), indicating that the monolayer adsorption was dominant.

### 3.3. Adsorption Characteristics of Pb onto the Beads

Figure 5 shows the adsorption isotherms of Pb onto the SCM beads (1/1/0.1, 1/1/0.3, and 1/1/0.5) and SP and the fitting for each model (the Langmuir, Freundlich, D-R, and DM models are shown in Figure 4a–d, respectively). The lowest Pb adsorption occurred in bead 1/1/0.1, and the highest adsorption amount occurred in bead 1/1/0.5. The Pb concentration (*q*_e_) adsorbed onto the beads significantly increased compared with that adsorbed onto SP and increased with the maghemite ratio in the beads (1/1/0.5 > 1/1/0.3 > 1/1/0.1). Table 5 shows the parameters derived from the regression analyses of the adsorption isotherm models. The R^2^ values for all model fitting results were greater than 0.618 in bead 1/1/0.3. The R^2^ value of the DM model was the highest of all the models.

In the Langmuir model, the maximum adsorption capacity (*q*_mL_) was 222.2, 200.8 and 232.8 mmol/kg for beads 1/1/0.1, 1/1/0.3, and 1/1/0.5, respectively, showing similar results regardless of the ratio. Comparing the *q*_mL_ values with that of SPs (41.8 mmol/kg), those of bead 1/1/0.1 increased by 5.3 times, bead 1/1/0.3 increased by 4.8 times, and bead 1/1/0.5 increased by 5.6 times. The adsorption affinity, *b*, also increased with the maghemite content. The R^2^ for the Langmuir model was in the range of 0.797–0.909. This makes it difficult to conclude whether the adsorption of Pb was a monolayer adsorption.

In Table 5, the Freundlich model showed high R^2^ values of over 0.93, which are higher than those of the Langmuir model, indicating the multi-layer adsorption of Pb onto the beads rather than monolayer adsorption. In addition, the K_F_ value—the affinity of the adsorption—increased with the SCM ratio from bead 1/1/0.1 to bead 1/1/0.5. The N value was in the range of 0.20–0.44, showing nonlinear adsorption.

The Pb adsorption onto the bead was also fitted with the D-R model. The R^2^ values ranged from 0.62 to 0.87, indicating relatively low accuracy. The q_mD_ values for beads 1/1/0.1, 1/1/0.3, and 1/1/0.5 were 151.9, 169.3, and 221.4 mmol/kg, respectively, and increased in proportion to the maghemite dose. The E values were all 8 kJ/mol or less, indicating that physical adsorption occurred.

The DM model can analyze the dominant model between the physical multi-layer adsorptions and the monolayer adsorption. Its R^2^ values were the highest (all above 0.950) of the models. This is because the beads have various adsorption sites composed of a composite component with a difference in adsorption affinity between the maghemite and SP. Unlike As adsorption, the K_pDM_ values in the model for Pb adsorption were high at 15.7, 14.9, and 9.1 for beads 1/1/0.1, 1/1/0.3, and 1/1/0.5, respectively, showing a decrease with increasing maghemite content. Conversely, the K_pDM_ value for the Pb adsorption onto SP was nearly 0. In addition, the q_mDM_ value was 66.0, 96.9, and 190.7 mmol/kg for beads 1/1/0.1, 1/1/0.3 and 1/1/0.5, respectively, increasing in proportion to the maghemite content.

### 3.4. pH Effect for the Maximum Adsorption Capacities of As and Pb onto the Beads and SP

Figure 6 shows the pH effect of As and Pb adsorption onto the beads, indicating that both As and Pb adsorption increased with pH. To better observe the effect of pH on As and Pb adsorption, the adsorption experiment was performed at a low concentration and interpreted using the Freundlich model. The magnitude of the K_F_ value in the model is related to the adsorption affinity. As shown in Figure 6 and Table 6, the K_F_ values increased with increasing pH, with 102.6 (at pH 7.0) > 100.3 (at pH 5.5) > 53.3 (at pH 4.0) for As adsorption and 512.7 (at pH 7.0) > 440.7 (at pH 5.5) > 394.4 (at pH 4.0) for Pb adsorption, indicating that the Pb adsorption affinity was higher overall than the As adsorption affinity.

### 3.5. Temperature Effect on the Adsorption of As and Pb onto the Beads and SP

A thermodynamics analysis was conducted for the adsorption of As and Pb onto the beads to estimate whether the reaction occurred spontaneously. The partition coefficient *K_p_* in the linear model was used as a thermodynamic parameter related to the Gibb’s free energy change, Δ*G*^0^ (kJ/mol) during adsorption. The changes in the enthalpy, Δ*H*^0^ (kJ/mol), and the entropy, Δ*S*^0^ (J/mol/K), were also calculated using the following equations:(2)$$\Delta G^{0}=-RT\ln K_{p}$$
(3)$$\ln K_{p}=\frac{\Delta S^{0}}{R}-\frac{\Delta H^{0}}{RT}$$
where *R* is the gas constant (8.314 J/K/mol), T is the absolute temperature in Kelvin, and *K_p_* is the partitional coefficient.

The effect of temperature on the adsorption of As and Pb onto the SCM beads is shown in Figure 7 and the thermodynamic parameters are presented in Table 7. Overall, the ln(K*_p_*) values for As and Pb adsorption increased as the temperature increased from 293 to 313 K. As shown in Table 7, the ΔH° value for As is higher than that for Pb, indicating that the As adsorption is a more endothermic reaction than the Pb adsorption. The ΔS° values for As and Pb are 133.1 and 79.16, respectively. The positive values indicate that the adsorption process was accompanied by structural changes in the adsorbent and adsorbate [20]. The negative values of the change in the free energy (ΔG°) indicated that the adsorption isotherms of both As and Pb were spontaneous in nature and that the value for Pb was lower than that for As, causing more active Pb adsorption than As adsorption onto the beads. However, the changes in the enthalpy (ΔH°) were negative, indicating an endothermic reaction. Overall, the reaction ratios of these adsorptions increased with temperature.

## 4. Conclusions

SCM composite beads were prepared with different mass ratios of 1/1/0.1, 1/1/0.3, and 1/1/0.5 of SP/chitosan/maghemite, respectively, for the adsorption of As and Pb. The composite beads had maximum uptakes of 1.83 and 50.0 mmol/kg for As and Pb, respectively. The optimum mass ratio of SP, chitosan, and maghemite in the bead for the maximum adsorption capacity was 1/1/0.5. The experimental data were analyzed using several adsorption isotherm models. From the results of the model fitting, the beads had multiple adsorption sites with different affinities resulting from the inclusion of maghemite compared with SP and the DM model was the best model to describe these differences. The adsorption also depended on the pH and temperature. The thermodynamic study indicated that the adsorption reactions of As and Pb onto the beads are endothermic and spontaneous resulting from the large increase in the entropy change. In conclusion, beads could be used as good effective adsorbents for As and Pb removal from water resources, including rivers and groundwater.

## Figures and Tables

**Figure 1 ijerph-18-08808-f001:**
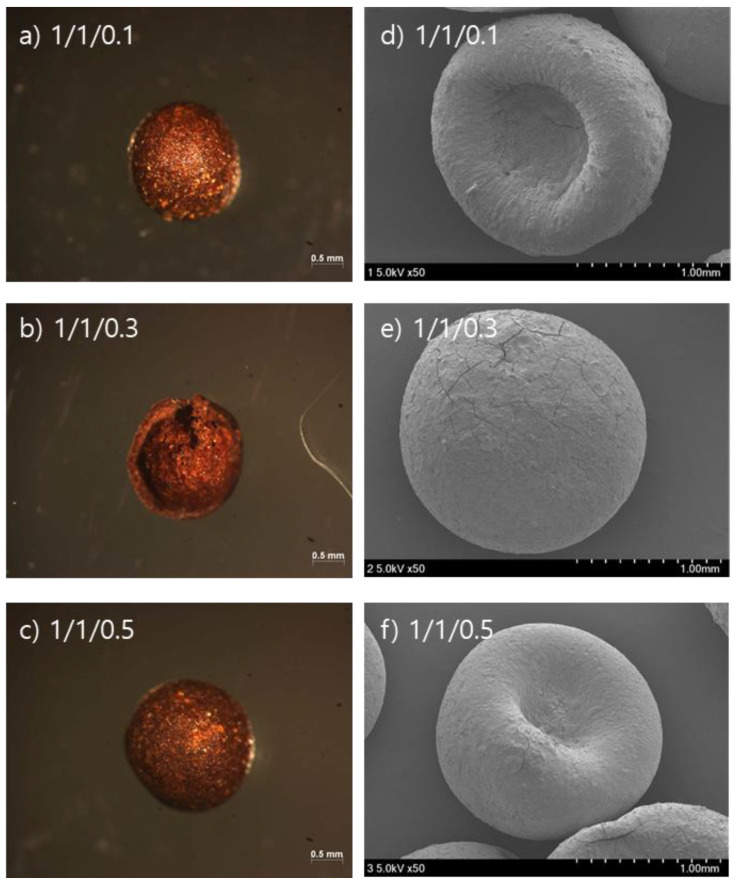
Microscopic (**a**–**c**) and SEM images (**d**–**f**) of SCM beads.

**Figure 2 ijerph-18-08808-f002:**
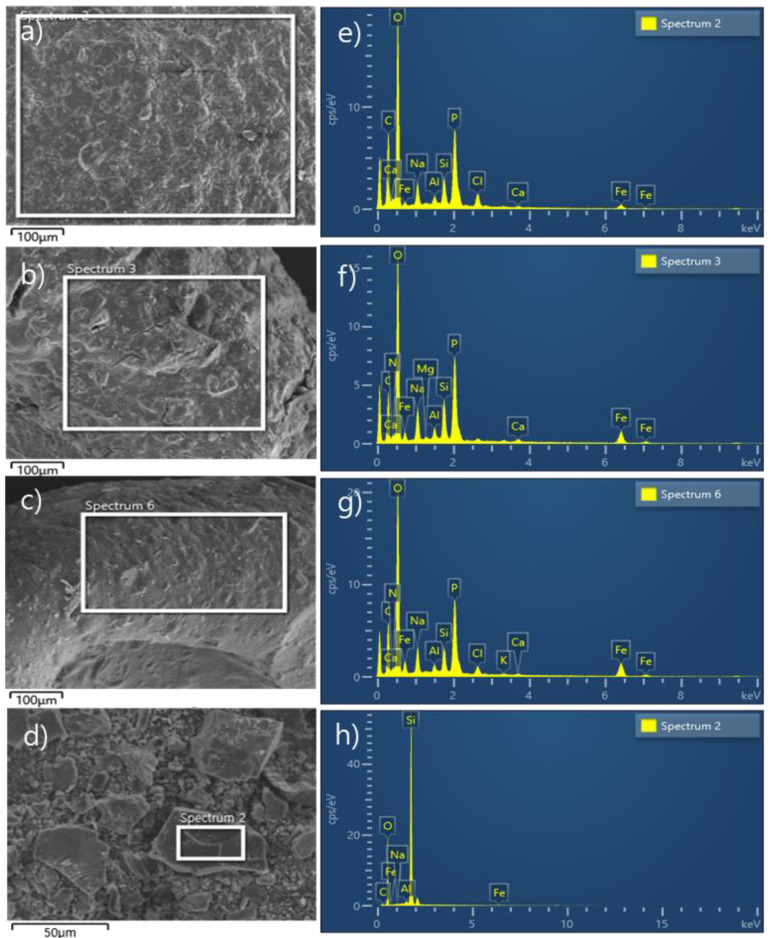
SEM images (**a**–**d**) and EDS spectra (**e**–**h**) of the SCM beads and SP: (**a**,**e**) bead 1/1/0.1; (**b**,**f**) bead 1/1/0.3; (**c**,**g**) bead 1/1/0.5; and (**d**,**h**) SP.

**Figure 3 ijerph-18-08808-f003:**
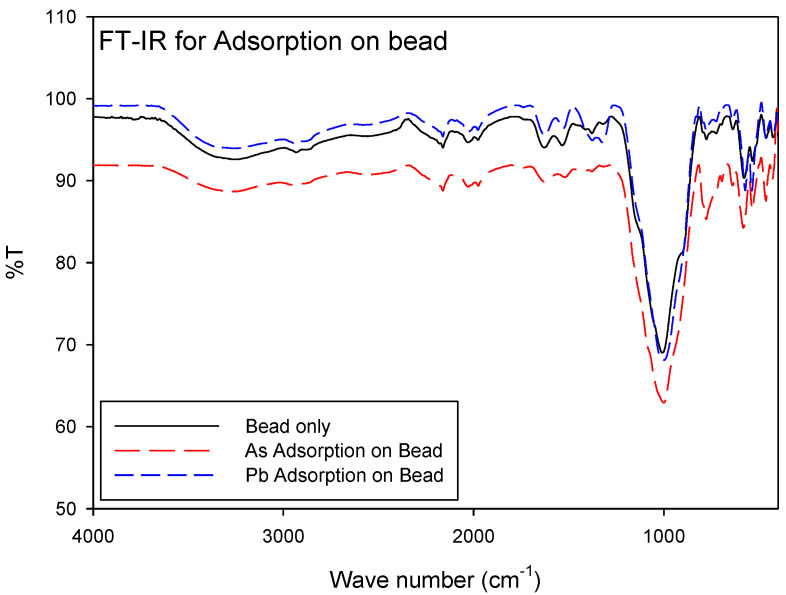
FT-IR spectra of the bead only; the As-loaded bead; and a Pb-loaded bead.

**Figure 4 ijerph-18-08808-f004:**
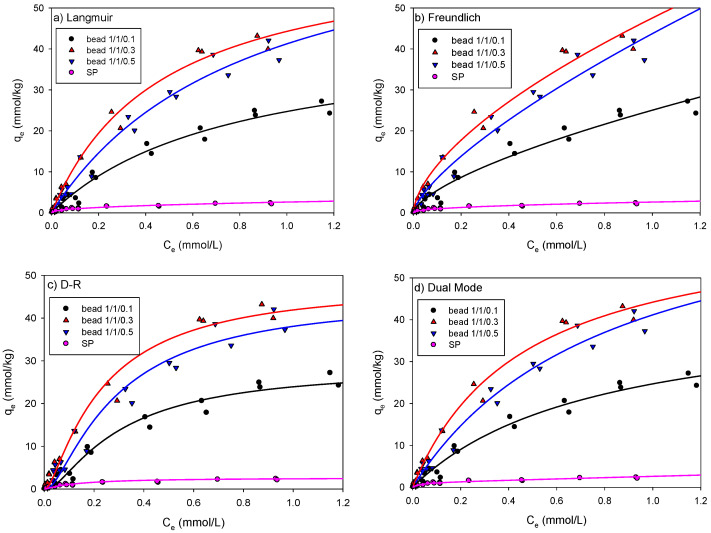
Nonlinear fitting of the (**a**) Langmuir; (**b**) Freundlich; (**c**) D-R; and (**d**) DM models for As adsorption onto the SCM beads and SP.

**Figure 5 ijerph-18-08808-f005:**
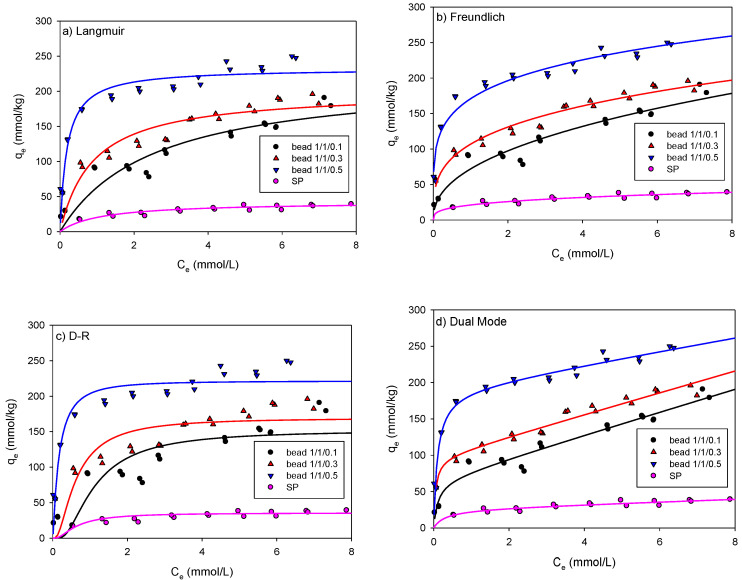
Nonlinear fitting of the (**a**) Langmuir; (**b**) Freundlich; (**c**) D-R; and (**d**) the dual mode models for Pb adsorption onto the beads and SP.

**Figure 6 ijerph-18-08808-f006:**
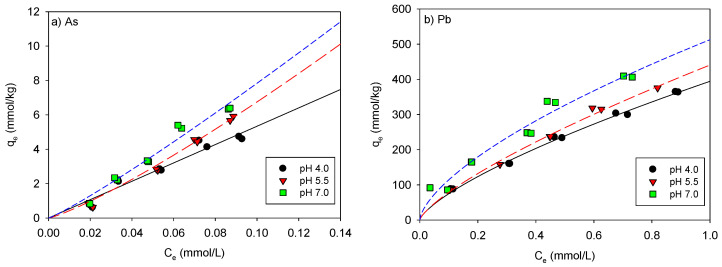
Effect of pH on the (**a**) As and (**b**) Pb adsorption fitted using the Freundlich model.

**Figure 7 ijerph-18-08808-f007:**
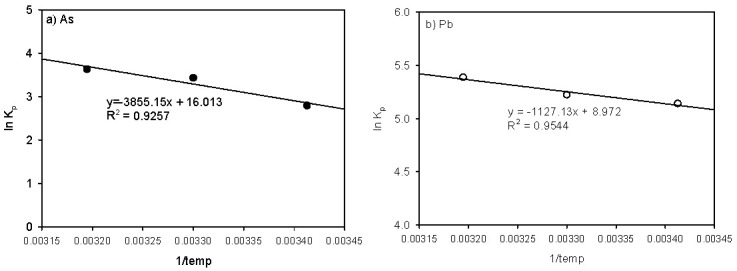
Temperature effects on (**a**) As and (**b**) Pb adsorption.

**Table 1 ijerph-18-08808-t001:** Properties of the SP/chitosan/maghemite (SCM) beads and SP.

		SCMs		SP [37]
	1/1/0.1	1/1/0.3	1/1/0/5	
pH	5.4	5.6	5.6	4.7
BET surface area (m^2^/g)	0.543	0.570	0.834	2.782
Pore volume (cm^3^/g)	0.00271	0.00397	0.00467	0.017
Pore size (nm)	19.98	27.86	22.42	24.83

**Table 2 ijerph-18-08808-t002:** EDS results for the SCMs and SP.

Element	Atomic % of SCMs			Atomic % of SP
	1/1/0.1	1/1/0.3	1/1/0/5	
C	45.34	35.47	37.56	8.52
N	-	4.39	4.88	-
O	45.79	46.55	45.92	53.88
Na	1.81	3.01	2.44	0.14
Mg	-	0.19	-	-
Al	0.39	0.61	0.41	0.36
Si	1.28	2.17	1.28	36.88
P	3.70	4.78	3.95	-
Cl	0.76	-	0.50	-
K	-	-	0.06	-
Ca	0.12	0.20	0.11	-
Fe	0.81	2.62	2.89	0.23

**Table 3 ijerph-18-08808-t003:** Adsorption isotherm models used in this study.

Model	Equation	Fitting Parameters
Langmuir	$q=q_{mL}bC/(1+bC)$	*q*_mL_ (mmol/kg) and *b* (L/mmol)
Freundlich	$q=K_{F}C^{N}$	*K*_F_ [(mmol/kg)/(mmol/L)*^N^*] and *N* (-)
Dubinin-Radushkevich (D-R) [40]	$\begin{array}{ll} q & =q_{mD}\exp(-\beta \varepsilon^{2})\\ & =q_{mD}\exp[-\beta {\left(RT\ln(1+1/C)\right)}^{2}] \end{array}$	*q*_mD_ (mmol/kg) and *β* (mol^2^/kJ^2^) $E=1/\sqrt{2\beta }$
Dual mode (DM) [41]	$q=K_{pDM}C+q_{mDM}b_{DM}C/(1+b_{DM}C)$	*K*_pDM_ (L/kg), *q*_mDM_ (mmol/kg), and *b*_DM_ (L/mmol)

**Table 4 ijerph-18-08808-t004:** Comparison of the isotherm model parameters for As.

Model	Parameter	SCM Bead			SP
		1/1/0.1	1/1/0.3	1/1/0.5	
Langmuir	*q*_mL_ (mmol/kg)	43.94	65.06	75.74	1.838
	b (L/mmol)	1.288	2.134	1.196	14.45
	R^2^	0.977	0.988	0.977	0.900
	SSE	42.43	41.18	104.7	0.134
	R_L_	0.017	0.007	0.011	0.036
Freundlich	*K*_F_ (mmol^1−*N*^ L*^N^*/kg)	25.06	47.56	43.74	4.072
	*N* (-)	0.662	0.613	0.717	0.572
	R^2^	0.966	0.979	0.967	0.899
	SSE	63.32	86.81	145.7	0.136
D-R	*q*_mD_ (mmol/kg)	28.07	47.22	44.30	1.883
	*β* (mol^2^/kJ^2^), ×10^−2^	5.432	4.051	5.143	1.570
	*E* (kJ/mol)	3.036	3.515	3.119	5.646
	R^2^	0.966	0.977	0.964	0.900
	SSE	63.29	92.33	158.9	0.134
DM	*K*_pDM_ (L/kg)	0.037	0.037	0.049	0.001
	*q*_mDM_ (mmol/kg)	43.49	64.62	74.90	1.834
	*b*_DM_ (L/mmol)	1.307	2.164	1.217	14.52
	R^2^	0.977	0.988	0.977	0.900
	SSE	42.46	47.21	104.7	0.134

**Table 5 ijerph-18-08808-t005:** Comparison of the isotherm model parameters for Pb.

Model	Parameter	SCM Beads			SP
		1/1/0.1	1/1/0.3	1/1/0.5	
Langmuir	*q*_mL_ (mmol/kg)	222.2	200.8	232.8	41.88
	b (L/mmol)	0.398	1.126	5.452	1.037
	R^2^	0.885	0.797	0.909	0.805
	SSE	5900	7042	5421	197.8
	R_L_	0.011	0.004	0.001	0.023
Freundlich	*K*_F_ (mmol^1−N^ L^N^/kg)	72.00	106.8	170.9	21.71
	*N* (-)	0.437	0.295	0.201	0.281
	R^2^	0.944	0.960	0.936	0.856
	SSE	2854	1386	3806	146.6
D-R	*q*_mD_ (mmol/kg)	151.9	169.3	221.4	35.86
	*β* (mol^2^/kJ^2^), ×10^−2^	29.40	12.90	2.831	13.10
	*E* (kJ/mol)	1.304	1.971	4.206	1.953
	R^2^	0.760	0.618	0.874	0.678
	SSE	12,228	13,245	7476	326.7
DM	*K*_pDM_ (L/kg)	15.70	14.94	9.108	4.62 × 10^−6^
	*q*_mDM_ (mmol/kg)	66.04	96.91	190.7	50.35
	*b*_DM_ (L/mmol)	9.069	21.06	10.08	0.464
	R^2^	0.958	0.969	0.950	0.679
	SSE	2151	1086	2993	325.7

**Table 6 ijerph-18-08808-t006:** Comparison of the *K*_F_ values in the Freundlich model.

pH	K_F_ (mmol^1−N^ L^N^/kg)	
	As	Pb
4.0	53.31	394.4
5.5	100.3	440.7
7.0	102.6	512.7

**Table 7 ijerph-18-08808-t007:** Thermodynamic parameters for the adsorption of As and Pb onto the beads.

	ΔH (kJ/mol)	ΔS (kJ/mol/K, ×10^3^)	ΔG (kJ/mol)		
			293 K	303 K	313 K
As	32.05	133.1	−6.955	−8.287	−9.618
Pb	8.457	79.16	−14.74	−15.53	−16.32

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
