# Peer review of "Adsorptive Removal of Arsenic and Lead by Stone Powder/Chitosan/Maghemite Composite Beads"

_ijerph, 2021, doi:10.3390/ijerph18168808_

Round 1

Reviewer 1 Report

The authors presented the removal of arsenic and lead by Stone Powder Chitosan-Maghemite Composite Bead.

This method shows the potential for effective removal of these elements in river and ground water.

The materials used in this paper, SCM beads, are described in detail in Table 1 and Future 2, and there is no problem.

The removal of As and Pb is shown in Table 3, 4. Future4, 5, with four models, Langmuir, Freundlich, D-R, and Dual mode, showing higher adsorption than stone powder (SP).

The results showed that the adsorption capacity of both Pb and As decreased at low pH. The adsorption of Pb was higher than that of As. (Fig. 6).

The data on the materials and results used in this paper are good, and the reviewer recommends that this article be published in Int. J. Environ. Res. Public Health.

Author Response

Thank you for your kind comments.

I revised the grammar mistakes in the manuscript by a native speaker.

Thanks.

Reviewer 2 Report

The authors report synthesized composite beads, named SCM beads, using stone powder, chitosan, and maghemite. The SCM beads are characterized and used as adsorbents for As and Pb from the contaminated water sample. The results reported are of great interest to the readers. The research is conducted with care and nicely presented. I would recommend the publication of the work if the followings concerns can be addressed properly. 

  1. In the introduction part, the motivation of this work on the synthesis of the SCM is to replace the clay with stone powder. Why it is better to replace the clay with stone powder? What are the scientific interests to conduct such research?
  2. Besides, the reference 31, 35, and 36 seems to be critical references for the current work but they are not latest works. What is the current status of the research in this field?   
  3. The authors compare the adsorption of As and Pb by SCM beads. Why do the authors compare these two heavy metal ions? What are the adsorption capacities of the SCM beads for other heavy metal ions?

Minor concerns:

  1. R2 is first introduced in the abstract. In the main text, R2 is analyzed without explaining its meaning.
  2. some typo of the manuscript is caught. 
    1. line 92, PH 2;
    2. line 187, broken parentheses;

Author Response

Response to Reviewer 2

The authors report synthesized composite beads, named SCM beads, using stone powder, chitosan, and maghemite. The SCM beads are characterized and used as adsorbents for As and Pb from the contaminated water sample. The results reported are of great interest to the readers. The research is conducted with care and nicely presented. I would recommend the publication of the work if the followings concerns can be addressed properly. 

1. In the introduction part, the motivation of this work on the synthesis of the SCM is to replace the clay with stone powder. Why it is better to replace the clay with stone powder? What are the scientific interests to conduct such research?

- A large amount of stone powder is generated every year, but most of it is thrown away as specific waste. During rainfall, untreated stone powder has the potential to flow into rivers, lakes, wetlands, etc. and cause environmental pollution. Therefore, if the stone powder is recycled, the amount of waste generated can be reduced and environmental pollution can be prevented. In this study, it is proposed to use the stone powder with similar properties as a bead support instead of clay.

- This study has scientific significance in that it developed an environmental material that treats toxic substances such as arsenic and lead using waste-stone powder.

2. Besides, the reference 31, 35, and 36 seems to be critical references for the current work but they are not latest works. What is the current status of the research in this field?

- The adsorption treatment of contaminants using magnetic composite beads has been extensively studied. Recently, a lot of studies on adsorption of radioactivity (Liu et al. 2019), heavy metals (Dolgormaa et al. 2020), emerging pollutants, etc. onto the various types of composite beads, e.g., nanosized alginate composite beads (Aziz et al. 2019).

Liu L, Yang W, Gu D, Zhao X, Pan Q. In situ preparation of chitosan/ZIF-8 composite beads for highly efficient removal of U(VI). Front. Chem. (2019).

Dolgormaa M, Tamiraa G, Ariunzul N, Koichiro S, Ochirkjuyag B. Pb(II) adsorption of composite alginate beads containing mesoporous natural zeolite. J. Nanosci. Nanotechnol. 20(8) (2020)

Aziz F, Achaby ME, Lissaneddine A, Aziz K, Ouazzani N, Mamouni R, Mandi L. Composites with alginate beads: A novel design of nano-adsorbents impregnation for large-scale continuous flow wastewater treatment pilots. Saudi J. Biol. Sci. 27(10), 2499-2508 (2020).

3. The authors compare the adsorption of As and Pb by SCM beads. Why do the authors compare these two heavy metal ions? What are the adsorption capacities of the SCM beads for other heavy metal ions?

- In this study, arsenic (As) and lead (Pb) were selected as target materials. According to a report from the Agency for Toxic Substances and Disease Registry (ATSDR) in the USA, As and Pb are the most prioritized substances, given their frequency, toxicity, and potential for human exposure. As and Pb are considered as a representative metalloid and a representative heavy metal, respectively. In addition, their main exposure pathway is groundwater, one of the most important water supplies in the world. For the above reasons, As and Pb were selected as the target materials in this study.

- We have not conducted any experiment for the adsorption of other heavy metals onto the beads. Therefore, we have not had any experimental data of adsorption capacities of the SCM beads for other heavy metals. We have several additional experimental plans for adsorption of radioactive, other heavy metals, emerging contaminants and so on.

Minor concerns:

1. Ris first introduced in the abstract. In the main text, R2 is analyzed without explaining its meaning.

- R2 is the determinant coefficient. The definition of R2 is added into the revised manuscript (line 206)

2. some typo of the manuscript is caught. 

   1. line 92, PH 2;

- PH 2 was revised to be “a pH of 2” in the revised manuscript (line 94)

    2. line 187, broken parentheses;

- To prevent the parentheses in the table 3 from being broken, the width of the cell was increased (line 193 in the revised manuscript)

The revised manuscript is also attached below.

Reviewer 3 Report

Its well organized. There has no comments. 

Author Response

(The authors gave the same response as above.)

Round 2

Reviewer 2 Report

I have no further suggestions on the manuscript.